# Improvements in Visual Aspects and Chemical, Techno-Functional and Rheological Characteristics of Cricket Powder (*Gryllus bimaculatus*) by Solvent Treatment for Food Utilization

**DOI:** 10.3390/foods12071422

**Published:** 2023-03-27

**Authors:** Barbara Catalano Damasceno, Mitsutoshi Nakajima, Noamane Taarji, Isao Kobayashi, Sosaku Ichikawa, Marcos A. Neves

**Affiliations:** 1Graduate School of Science and Technology, University of Tsukuba, Tsukuba 305-8577, Japan; 2Faculty of Life and Environmental Sciences, University of Tsukuba, Tsukuba 305-8577, Japan; 3College of Sustainable Agriculture and Environmental Sciences, AgroBiosciences Program, Mohamed 6 Polytechnic University, Ben Guerir 43150, Morocco; 4Institute of Food Research, NARO, Tsukuba 305-8642, Japan

**Keywords:** *Gryllus bimaculatus*, alternative proteins, insects, solvent treatment

## Abstract

This study aimed to improve the visual aspects and chemical, techno-functional and rheological characteristics of *Gryllus bimaculatus* cricket powder through the use of different solvents, with the objective of using it as a protein source in food production. Four treatments (pH 5 aqueous solution, ethanol 20%, ethanol 99.5%, and hexane) were applied to the powder, and analyses were conducted to assess changes in the previously mentioned parameters. The results showed that the treatments led to an increase in protein concentration (from 55.4 to 72.5%) and a decrease in fat concentration (from 33.0 to 6.8%) in ethanol 99.5% treated powder, as well as a reduction in anti-nutritional compounds concentration, such as tannins (from 13.3 to 5.9 g/kg), in pH 5 treated powder, which is important for the nutritional value of the final product. The color of the powders was improved, being lighter after hexane and ethanol 99.5% treatments due to the removal of melanin with the defatting process. Flowability, water, and oil holding capacity were also improved in the defatted powders. All the results suggest that the main composition of the powder directly influences the analyzed parameters. These findings suggest that cricket powder treated with solvents can be used as a protein source in different food applications.

## 1. Introduction

The global population is increasing every day, and it is estimated that, in 2050, the global population will reach 9 billion people [1]. With that increase, the demand for animal-based protein is also expected to increase [1]. Although the consumption of dietary protein is crucial to human health [2], cattle are the main contributors to greenhouse gas (GHG) emissions, representing 62% of all emissions from the animal industry sector. According to the FAO [3], livestock production is responsible for elevated quantities of GHG emissions and 14.5 % of the world’s total emissions. Moreover, wastewater is also a problem for traditional livestock, since processing meat requires thousands of liters of water, e.g., for every kg of beef, 22 m^3^ of water is needed [4].

Alternative sources of protein have been researched over the years. Among them, edible insects are a new option as a source of protein, fat, and minerals, and are receiving attention all around the world [5]. The protein concentration of edible insects, for example, may vary by up to 48% in fresh basis weight [2], which is superior to the traditional sources of protein, such as chicken (18–20 g/100 g) [6], beef (19–26 g/100 g), tilapia (16–19 g/100 g), and shrimp (13–27 g/100 g) [2]. Insects such as crickets (*Orthoptera*) can meet or exceed the current essential amino acid requirements for adults, with the amino acid profile being comparable to that of beef, eggs, or milk [2]. The use of insects in the diet also has ecological and economic advantages when compared to other farming options, such as a reduction in GHG emissions and less wastewater when compared to traditional livestock [1].

Despite all these benefits, the consumption of edible insects has low acceptance, especially in countries where there is no recent history of entomophagy [7]. The use of insect powders is one option to increase the acceptance of this food. Insect powders are rich in proteins (60 g/100 g), fat (20 g/100 g), vitamins, and bioactive compounds, although the whole composition can change from one species to another based on farming conditions [8]. Several studies on insect powders have been performed due to their acceptability and long shelf life, which can last up to 18 months [9]. However, processing insects into powder is not enough to increase the acceptability and consumption of edible insects. The color of the powder can also increase the acceptability of this new food. According to Wadhera [10], the color of the food can influence the acceptance and ingestion of particular foods. Darker colors usually are related to bitter taste by consumers, since studies show that colors can affect flavor perception [10]. The color removal of insect powders could help to increase the acceptance of the final product.

In addition to the decolorization of the powder, other strategies to attract more consumers must be studied. One example is the use of 3D food printing (3DFP). This 3DFP, also known as additive manufacturing, is a new technology developed for the production of 3D objects made of food, and can produce different shapes and textures [11]. After the production of the 3DFP food, the printed object should keep its shape [12]. To ensure this, it is important to know the powder’s properties and flowability. 

However, even though the use of edible insect powder for 3D food printing is one option to increase the acceptability of this new type of food, it is important to characterize the insect powder and find ways to improve its characteristics. Powder treatments are one option for improving insect powder characteristics. Séré [13] showed an increase in protein concentration and an enhancement of functional parameters after the solvent treatment of insect powders, such as an improvement in digestibility, which was initially was around 82% and improved to 88% after powder treatment. Therefore, the main objective of this study is to characterize and improve the visual aspects and chemical, techno-functional and rheological properties of *Gryllus bimaculatus* powder using solvent treatments and foreseeing food utilization.

## 2. Materials and Methods

The *Gryllus bimaculatus* powder was obtained from Gryllus Inc. (Naruto, Japan). All the other reagents were purchased from FUJIFILM Wako Pure Chemical Corporation (Osaka, Japan) or Sigma Aldrich Chemical Co. (Tokyo, Japan).

### 2.1. Effect of the pH and Ethanol Concentration on Protein Behavior

A preliminary study was performed to optimize reductions in protein loss, using different pHs and ethanol concentrations.

The original cricket powder was mixed with Ultrapure water (Sartorius Arium^®^ Pro, Göttingen, Germany) (weight volume ratio 1:10), adjusted to different pH values (3.5, 4.0, 4.5, 5.0, 5.5, and 6.0) or aqueous ethanolic solutions with different ethanol concentrations (0, 20, 40, 60 and 80%), and kept under stirring for 2 h. Samples were then centrifuged for 30 min at 10,000× *g*. The supernatants were collected, and the protein concentration was evaluated using the Bradford method [14] and expressed as mg/mL. The optimum pH and ethanol concentration were then selected based on protein concentrations in the supernatant, indicating a minor loss of protein after the procedure. All the experiments were carried out in triplicate.

### 2.2. Solvent Treatment

For the preparation of the pH 5 aqueous solution, ethanol 20%, ethanol 99.5%, and hexane treatment, the protocols reported by Wang [15] were used with some modifications. For the pH 5.0 treatment, the sample was dispersed in Ultrapure water (Sartorius Arium^®^ Pro) (weight volume ratio 1:10) and the pH was adjusted to 5.0. After that, the solution was stirred at room temperature (RT), at about 20 °C for 2 h. The pH of the solution was adjusted every 30 min. Then, the solution was centrifuged for 30 min at 20,000× *g*. The supernatant was discarded, and the pellet was pre-dried in the fume hood overnight, frozen at −20 °C, and lyophilized (−80 °C at 5 Pa). The same procedure was adopted for the ethanol 20% and hexane treatments. For ethanol 99.5% treatment, the sample was dispersed in ethanol 99.5% (1:10 *w*/*v*) and was stirred for 2 h at 60 °C. The temperature of the solution was monitored throughout the experiment. The supernatant was discarded, and the sample was dried in the fume hood overnight, followed by lyophilization (−20 °C at 5 Pa). The increase in the temperature and use of ethanol 99.5% were applied to decolorize and improve other parameters in the powder, and for comparison with other treatments. After the drying process, all the samples were sieved (Tokyo Screen Co. Ltd., Tokyo, Japan) (aperture 500 µm) and stored at −20 °C.

### 2.3. Color Difference after Treatments

The color of the original and treated powders was evaluated in the CIE LAB system, using a CM-5 spectrophotometer (Konica Minolta, Osaka, Japan). The evaluated parameters were lightness (*L**), redness (*a**), and yellowness (*b**). The illuminant used was D65, with a standard observer of 10°, and the measurement time was 1 s. All the experiments were conducted in triplicate.

### 2.4. Moisture Content

The moisture content was evaluated following the AOAC (925.09) [16] method. A total of 1 g of sample (wet basis) was precisely weighed on a dry pre-weighed aluminum pan and dried in an oven at 105 °C for 24 h. After that, the samples were cooled at RT in a desiccator and then weighed. The moisture content was calculated from the weight difference due to water loss. All the experiments were carried out in triplicate.

### 2.5. Ash Concentration

The ash concentration was evaluated following the AOAC method (942.05) [17] with modifications. In summary, 2 g of the sample (wet basis) was placed in ceramic crucibles and dried in an oven at 105 °C for 12 h. After that, the samples were ignited in a furnace oven at 550 °C for 24 h. Then, the samples were cooled in the desiccator and weighed after reaching RT. The ash concentration (g/100 g) in the dry basis was then calculated. All the experiments were carried out in triplicate.

### 2.6. Fat Concentration

The fat concentration was determined following the AOAC method (991.36) [16], with some modifications. Fat concentration was calculated by extracting the fat of the samples (from 7 to 15 g of sample in wet basis) in a Soxhlet extractor using hexane (300 mL) for 24 h. After that, the solvent was evaporated using a rotary evaporator (under 50 °C and 49 kPa), and the fat was weighed. The fat concentration (g/100 g) was calculated. All the experiments were carried out in triplicate. Hexane was selected as the best solvent, after comparison with the extraction yields obtained using petroleum ether or hexane. 

### 2.7. Protein Concentration

The nitrogen concentration of the samples was determined using an Organic Element Analyzer (UNICUBE, Elementary Ltd., Yokohama, Japan) followed by the calculation of protein concentration (g/100 g) using nitrogen to protein conversion (factor 6.25) [18]. The nitrogen present in chitin was discounted from the total nitrogen before conversion. The experiments were carried out in triplicate.

### 2.8. Chitin Concentration

The chitin concentration was determined following the method of Liu [19], with some adaptations. The samples were defatted using 1:20 *w*/*v* with hexane for 24 h at RT under stirring. After that, the solution was centrifuged, and the supernatant was discarded. The pellet was dried in the fume hood overnight. In the sequence, the minerals present in the powder were removed using 1M HCl (1:20 *w*/*v*) at 80 °C for 50 min. Then, the powder was filtered using a glass fiber filter, under vacuum. For protein removal, after the filtration of the powder, the precipitate was treated with 1M NaOH (1:20 *w*/*v*) at 100 °C for 20 h and then filtered again. The powder after filtration was placed in crucibles (dry pre-weighed) and set to dry in the oven at 105 °C for 3 days (or until the weight was constant). The chitin concentration was determined using the following equation:(1)$$Chitin\left(\frac{g}{100\;g}\right)=\frac{weight\;of\;dry\;matter-weight\;of\;the\;crucibles}{sample\;weight}$$

### 2.9. Total Phenolic Concentration

To determine the total phenolic concentration, the samples were first extracted. The extraction was carried out with 250 mg (wet basis) of samples mixed with 2 mL of methanol water solution (50:50 *v*/*v*) and vortexed for 30 min at RT. The suspensions were centrifuged for 30 min at 10,000× *g*. The supernatant was placed in a different tube, and the pellet was re-extracted using a methanol/water solution (80:20 *v*/*v*) for 30 min while being vortexed. Then, the suspensions were centrifuged for 30 min at 10,000× *g* and the supernatant was combined with the previous one. The extract was filtered using a 0.45 µm PTFE syringe filter and completed to 5 mL with Ultrapure water (Sartorius Arium^®^ Pro). Then, the extracts were diluted 3 times with Milli Q. A total of 0.5 mL of the diluted extract was mixed with 2.5 mL Folin–Ciocalteu reagent (1N) and 2 mL of 7.5% sodium carbonate aqueous solution. The mixture was heated at 45 °C for 15 min in the dark. Optical density was taken using a spectrophotometer (V530, JASCO Co., Tokyo, Japan) at 765 nm. The total phenolic concentration was estimated using a standard curve of gallic acid. The standard curve was prepared using gallic acid in concentrations of 20, 40, 60, 80, 100 and 120 µg/mL (R^2^ = 0.9906). The total phenolic concentration was expressed in g/kg gallic acid equivalent. All the experiments were carried out in triplicate.

### 2.10. Osborne Classification

The classification of the protein fractions in the original cricket powder was determined following the Kumar method [20] with modifications. For the extraction of the water-soluble protein (albumin), 3 g (wet basis) of cricket powder was dispersed in 30 mL of Ultrapure water (Sartorius Arium^®^ Pro). The solution was stirred for 2 h at RT followed by centrifugation at 10,000× *g* for 20 min. The supernatant was collected and frozen in a Petri dish. After that, the pellet was resuspended using 30 mL of NaCl 1 M solution to extract the salt-soluble proteins (globulin). The same process was repeated as above, followed by extractions with the pellet in alkaline conditions (pH 12) to extract glutelin, and then extraction with ethanol 99.5% to evaluate prolamin content. All the supernatants were frozen except for the ethanol solution that was evaporated in the fume hood, and lyophilized (−20 °C at 5 Pa). The protein concentration of the samples was determined by Nitrogen Elemental Analysis and by multiplying the N content by 6.25. All the experiments were carried out in triplicate.

### 2.11. Tannins Concentration

The tannins concentration was determined following the Tandon method [21]. In summary, for sample extraction, 400 mg of the sample (wet basis) was diluted in 20 mL of 70% aqueous acetone solution and kept under stirring for 2 h at RT. Then, the samples were filtered in a vacuum filter using a Whatman filter paper No. 1 and the volume of the extraction was made up to 20 mL using Ultrapure water (Sartorius Arium^®^ Pro). After that, 100 µL of the extract was transferred to a glass tube and 900 µL of Ultrapure water (Sartorius Arium^®^ Pro) was added. Then, 500 µL of Folin–Ciocalteu reagent (1 N) and 2.5 mL of Na_2_CO_3_ 20% (*w*/*w*) aqueous solution were mixed into the tube. The solutions were kept at RT in the dark for 40 min. Optical density was determined using a spectrophotometer (V530, JASCO Co., Tokyo, Japan) at 725 nm. The concentration of tannins was estimated using a standard curve of tannic acid. The standard curve was prepared using tannic acid at concentrations of 10, 20, 30, 40, 50 and 60 µg/mL (R^2^ = 0.9931). The tannins concentration (g/kg) was expressed as tannic acid equivalent. All the experiments were carried out in triplicate.

### 2.12. Phytates Concentration

The phytates concentration was determined according to the method described by Raboy [22]. In brief, 500 mg of samples (wet basis) were mixed with 5 mL of 0.2 M HCl: 5% Na_2_SO_4_ and extracted overnight at 4 °C. Then, the samples were centrifuged for 20 min at 10,000× *g* and 250 µL of the supernatant was allocated to a glass tube with 2.25 mL of 0.2 M HCl:5% Na_2_SO_4_ and 5 mL of iron solution (0.2 g of (NH_4_)Fe(SO_4_)2·12H_2_O in diluted in 1 L of 0.2 M HCl) and placed in a boiling water bath for 30 min. Following this, the samples were cooled at RT and centrifuged for 20 min at 4500× *g*; 1 mL of the supernatant was mixed with 1.5 mL of HL reagent (1 g of 2,2′ bipyridine, 1 mL thioglycolic acid dissolved in 100 mL Ultrapure water (Sartorius Arium^®^ Pro)). The absorbance of the samples was read using a spectrophotometer (V530, JASCO Co., Tokyo, Japan) at 519 nm. The time between the addition of the HL reagent to the samples and the readings was 1 min. The phytates concentration (mg/kg) was determined using a phytic acid standard curve. The standard curve was prepared using phytic acid at concentrations of 0.625, 0.8, 1, 2, 4, 8 and 10 µg/mL. All the experiments were carried out in triplicate.

### 2.13. pH

For determining the pH, 1 g of the samples (wet basis) was dispersed in 10 mL of Ultrapure water (Sartorius Arium^®^ Pro) and kept under stirring for 5 min. Then, the samples were left at RT for 10 min to allow the powder to settle. The pH was read using a Mettler Toledo Seven Easy pH meter. All the experiments were carried out in triplicate.

### 2.14. Zeta Potential

To determine the zeta potential, the samples (1 g in wet basis) were dispersed in Ultrapure water (Sartorius Arium^®^ Pro) (10 mL) and then diluted 10 times with Ultrapure (Sartorius Arium^®^ Pro) water again. The sample was allocated to capillary cells and the zeta potential was determined using a Zetasizer Nano ZS (Malvern Instruments Ltd., Worcestershire, UK). All the experiments were carried out in triplicate.

### 2.15. Water-Holding Capacity (WHC) and Oil-Holding Capacity (OHC)

To determine the WHC, 1 g of samples (wet basis) were dispersed in 25 mL of Ultrapure water (Sartorius Arium^®^ Pro) in a pre-weighed centrifuge tube, and homogenized for 3 min. The dispersions were kept for 30 min at RT to allow the absorption of water. Then, the samples were centrifuged for 15 min at 10,000× *g*. The supernatant was discarded, and the tubes were kept upside-down to remove all the water for 1 min. Then, the tubes with the pellets were weighed again and the WHC was determined by the difference between the initial and final weights. The same procedure was adopted to determine OHC, using soybean oil (FUJIFILM Wako Pure Chemical) instead of water. All the experiments were carried out in triplicate.

### 2.16. Flowability 

The powder properties were evaluated and measured using Powder Tester (Hosokawa Powder Tester PT-X, Osaka, Japan). The flowability was evaluated through the measurements of the following parameters: angle of repose and cohesion. All the experiments were carried out in triplicate.

### 2.17. Viscosity 

The viscosity and shear stress in the function of shear rate were determined using a DV2T rheometer (Brookfield, Middleboro, MA, USA) with a number 27 spindle. Pastes containing 30 % (*w*/*w*) powder of each treated sample fully mixed in Ultrapure water (Sartorius Arium^®^ Pro) were prepared. All the experiments were carried out in triplicate.

### 2.18. Total Recovery and Recoveries of Each Compound

The total recovery after the treatment of the powders was determined using the following equation: (2)$$Total\;powder\;recovery\left(\%\right)=\left(\frac{final\;weight}{initial\;weight}\right)\times 100$$

The recoveries of protein, fat, chitin, phenolic compounds, tannins and phytates were determined using the following equation: (3)$$Recovery\;of\;the\;compound\;(\%)=\left(\frac{final\;weight\;of\;the\;compound\;in\;the\;recovered\;powder}{initial\;weight\;of\;the\;compound\;in\;the\;powder}\right)\times 100$$

The final weight of the compound in the recovered powder was determined from total recovery and the compound concentration in the recovered powder.

### 2.19. Statistical Analysis

For statistical analysis, all the measurements were performed in triplicates. The statistical test used for analysis was the one-way analysis of variance (ANOVA) with Holm–Sidak as a post hoc. To evaluate the correlation between the powder composition and powder properties, Pearson’s Correlation test was performed. All the analyses were performed using SPSS statistics software, version 29.0 (IBM Corp., Armonk, NY, USA). The results were expressed as mean and standard deviation. Significant difference was considered at *p <* 0.05.

## 3. Results and Discussion

### 3.1. Effect of the pH and Ethanol Concentration on Protein Behavior

The effects of the pH conditions and different ethanol concentrations on protein behavior are shown in Figure 1 and Figure 2.

It is possible to observe that, in low pH conditions (pH 3.5), the protein concentration in the supernatant is higher (1.0 mg/mL), indicating a possible protein loss from the powder. If this level of pH is chosen as a solvent treatment, there will be greater protein loss in the final product. Moreover, protein solubility in aqueous solutions can change according to the pH of the solution [23]. When protein concentration was analyzed at pH 5 in this study, low protein concentration (*p* < 0.05) was observed in the supernatant (0.26 mg/mL), indicating that this level of pH might be the best for use as an aqueous solvent for treating the cricket powder, since most of the protein concentration will remain in the powder and will not be transferred to the supernatant. Consistent with our data, a similar result was found by Pelegrine et al. [24] when the solubility of whey protein was analyzed as a function of pH, showing that the low protein concentration in the supernatant was found in pH 4.5, and lower pHs led to a higher protein loss in the supernatant.

Zidani et al. [25] also obtained a similar result when evaluating yeast protein solubility as a function of pH, with a minor protein concentration in the supernatant at pH 4. 

When we analyzed the protein concentration in the supernatant as a function of the ethanol concentration, ethanol 20% was the best option for use as a treatment since it showed the lowest concentration of protein in the supernatant. With these data, the ethanol 20% condition was chosen as the solvent used for treatment in the following experiments. The Bradford technique, used in this experiment to evaluate the protein loss, is only applicable to aqueous solutions; it was not possible to use it to evaluate the protein loss in the ethanol 99.5% condition. Therefore, we also chose to use Osborne Classification, and ethanol 99.5% was chosen for use as a treatment once it was efficient in removing fat from the sample and in the decolorization of the powder, as discussed further in Section 3.5.1. 

### 3.2. Osborne’s Classification of Proteins

The original powder was analyzed based on the Osborne Classification (Figure 3), and proteins were classified according to their solubility [26]. Our study revealed that in the *Gryllus bimaculatus* powder, 56.41% of the proteins were albumin (water soluble), 32.48% glutelin (alkaline pH soluble), 7.90% prolamin (ethanol soluble) and 3.21% globulin (saline soluble). In contrast to our results, Stone [27] showed that in *Acheta domesticus* protein concentrate, the most abundant proteins were albumin, with 31.5%, followed by globulin, prolamin and glutelin. This difference might be because the cricket analyzed in this study belongs to a different species. In another study of defatted cricket powder (*Acheta domesticus*), the second most abundant protein was glutelin, a result consistent with the one found in this study [28].

### 3.3. Total Powder Recovery

The total powder recoveries are presented in Table 1. The recoveries were between 63.3% (ethanol 99.5%-treated powder) and 79.1% (pH 5-treated powder). Among all the samples, pH 5 had the highest recovery (*p <* 0.05) (79.1%) when compared with the other samples, as the removal of fat or other major compounds from the powder was reduced. Consistent with our results, Wang et al. [15] reported similar recoveries by treating soybean powders with ethanol and acidic wash to produce protein concentrates. Ndiritu et al. [28] showed that, when the powder of *Acheta domesticus* was treated with hexane, the recovery was 66.3%, which was similar to the one found in this study. Our data showed that it is possible to speculate that this recovery is related to the defatting from the original cricket powder during the treatment.

### 3.4. Chemical Characterization of the Powders 

#### 3.4.1. Proximate Analyses

The results for proximate analyses of all the cricket powders and recoveries are shown in Table 1. The results revealed that, in the original powder, protein was the principal component of cricket powder (55.4%), followed by fat (33.0%) and chitin (7.3%). According to Liu [29], the concentration of protein in edible insects usually ranges between 30 and 70% on a dry basis.

Our data showed that protein concentration was within the expected range. For that reason, edible insects, including *Gryllus bimaculatus*, can be classified as “high protein food” [30]. The concentration of fat in edible insects can vary from 12 to 32%, and the chitin concentration can be around 6% in crickets [31]. In the present study, *Gryllus bimaculatus* powder had a high amount of fat, which can be disadvantageous. According to Udomsil et al. [32], the high fat concentration can shorten the shelf life of the powder as the fat can oxidize easily.

Thus, the concentration of chitin found in this study was very similar to the one reported by a previous study using *Acheta domesticus* powder [31], showing that in both species the chitin concentration does not significantly vary. 

After the treatments, the powders showed a significant increase in protein (from 55.4 to 72.5%) (*p <* 0.05) and chitin concentration (from 7.3 to 9.8%) (*p <* 0.05), and an expressive decrease in fat concentration (from 33.0 to 6.8%) (*p <* 0.05), The decrease in fat concentration is related to the polarity of the solvent used; non-polar solvents such as hexane will extract the fat from the samples. The recoveries of these compounds follow the same pattern. The hexane-treated powder showed a protein recovery of 83%, the highest reported recovery, followed by ethanol 99.5%, pH 5, and ethanol 20% treated powder. The fat recoveries ranged between 13.0 (ethanol 99.5% treated powder) and 70.3% (pH 5 treated powder). This is due to the fact that the hexane and ethanol 99.5% removed the fat from the sample more efficiently. After removing the fat, the concentration of other compounds was expected to increase. Therefore, the recovery of chitin in the samples ranged from 82.3 to 93.5%.

The increase in protein concentration can be observed in all the treated powders, especially in the ethanol 99.5% treatment due to the fat reduction of the powders. The defatting also resulted in a decrease in calories per portion. Gravel et al. reported the same pattern when *Tenebrio molitor* powder was defatted using hexane [33]. In the same study, after the defatting process, the concentration of proteins increased from 38.6 to 59.1% when the fat concentration was reduced from 28 to 0.4% and the chitin concentration was increased [33]. Our data revealed a more significant increase in protein concentration in *Gryllus bimaculatus* than the data shown in *Tenebrio molitor*. This difference in protein concentration can be related not only to the species analyzed in both studies, but also to the different methodologies applied in each study.

The ash concentration and recovery are shown in Table 1. The ash concentration ranged from 1.5 to 4.1 g/100 g. The ash concentration of pH5-treated powder (1.5%) was significantly lower (*p <* 0.05) when compared to the original powder (3.2%), but no significant reduction was observed in the ethanol 20% treated powder. The recovery of ash was higher in the presence of non-polar solvent treatments (85.5%) when compared with the samples treated with aqueous solvents (polar solvents). Ash concentration is an indicator of mineral concentration in foods [34]. This decrease indicates a possible reduction in the mineral concentration of the samples after pH5 and ethanol 20% treatments, which was expected since those two treatments used aqueous solvents and minerals are water-soluble [34,35]. 

#### 3.4.2. Total Phenolic Compounds and Antinutritional Compounds 

The total phenolic concentration and recovery of *Gryllus bimaculatus* and the treated powders are also shown in Table 1. The original powder revealed 6.0 g/kg of total phenolic concentration. However, this value increased significantly to 7.7 g/kg (*p <* 0.05) after treatment with ethanol 99.5% due to the removal of fat from the powder, which led to the concentration of the remaining compounds and increased the concentration of these total phenolic compounds. In treatments with aqueous solvents, we observed a significant loss of total phenolic compounds when compared to the original powder (*p <* 0.05), from 6.0 to 4.3 g/kg, indicating a possible predominance in the water-soluble total phenolic compounds in the powders. Total phenolic compounds have several bio-functions, such as antioxidant and inflammatory functions [8]. Nino et al. [8] showed that the values of total phenolic compounds in the original powder of different edible insects can change from 0.8 to 36 g/kg, indicating that the analyzed powders had high concentrations of total phenolic compounds. Interestingly, Navarro del Hierro et al. [36] showed that *Acheta domesticus*, for example, had approximately 5.0 g/kg of total phenolic compounds, which is consistent with the results found in the original *Gryllus bimaculatus* powder in our study. The recovery after treatments was consistent with the obtained concentrations. The highest recovery of total phenolic compounds was 82%, in the ethanol 99.5% treated powder, which was significantly higher than that obtained in the powders treated with aqueous solvents. 

The concentration and recoveries of anti-nutritional compounds, such as tannins and phytates, in the original and treated cricket powders, are shown in Table 1. Our study showed that the concentration of tannins in the original powder was 13.3 g/kg. In the powders treated with aqueous solvents, it was possible to notice a reduction to 5.9 g/kg of this undesirable compound concentration (*p <* 0.05) and recovery (55.6%). This reduction might occur because tannins are water-soluble compounds and were eliminated during the treatment [37]. Antinutritional compounds are defined as components present in food that can reduce nutrient utilization or absorption [38]. Since most edible insects are herbivores, they can accumulate these compounds [39]. According to Meyer-Rochow et al. [39], the tannin concentrations in crickets and grasshoppers ranged from 9.0 to 10.5 g/kg. Our data showed that aqueous treatment significantly reduced the tannins compound from 13.0 to 5.9 g/kg, which greatly improved the nutritional value of the final product, since these compounds can interfere with the nutrient absorption [38].

As for phytates, in the analyzed cricket powder the concentration of phytates found was 9.30 mg/kg in the original powder. When treated with pH 5 aqueous solution, a significant reduction (*p <* 0.05) was observed from 9.30 to 8.75 mg/kg when compared to the original powder. Regardless of the reduction in the concentration, the recovery of phytates was higher in the pH 5 powder. This might be due to the higher recovery rate of this powder.

Since phytate is water soluble [40], during the aqueous treatment this compound was partially eliminated. Usually, the concentration of phytates in crickets and grasshoppers ranges from 11.0 to 31.5 g/kg [39], which might indicate that the *Gryllus bimaculatus* powder has a low phytate concentration.

### 3.5. Visual Aspects, Techno-Functional and Rheological Characterization of Powders

#### 3.5.1. Color Change after Powder Treatment

Figure 4 shows pictures of the powder’s color before and after treatments. The color of the powders treated with pH 5 and ethanol 20% showed no improvement. After treatment with ethanol 99.5% and hexane, the powders showed a lighter color when compared to the other powders.

Figure 5 shows the reflected color value of the powders. The color parameters for the original powder and the treated ones are expressed as *L**, *a** and *b**. *L** expresses the lightness of the sample, while *a** is related to a red color, and *b** a yellow color [41]. In the original powder, the *L** parameter was 50.4. However, this value significantly increased (*p <* 0.05) to 74.2 due to the defatting process in samples treated with ethanol 99.5% and hexane, resulting in a lighter color of the samples. The brownish color present in the *Gryllus bimaculatus* powder is associated with melanin expression in this insect [28]. Due to the fact that melanin is a lipophilic compound present in the skin of animals [42], the removal of fat from the samples also removes melanin. This explains the lighter color result in hexane and ethanol 99.5% treated powders. For *a** and *b** it is possible to observe a reduction of these parameters (*p <* 0.05). For *a**, the reduction is from 3.7 in the original powder to 1.3 in the hexane-treated powder. For *b**, the reduction occurs from 17.9 (original powder) to 10.0 (hexane-treated powder). A similar pattern was observed by Delgado-García et al. [43] when defatted amaranth flour was compared with full-fat flour. Due to the removal of fat, in lipophilic compounds such as pigments were removed from the sample; therefore, *a** decreased from 2.3 to 1.1 and *b** from 19.2 to 15.8 [43].

In addition to nutritional values, color is an important parameter in the food industry. It is one of the first properties evaluated by the final consumers [44]. Ndiritu et al. [28] reported that, after treatment with hexane in *Acheta domesticus*, the *L** improved from 58.53 to 65.50. This difference might be attributed to the difference in the analyzed species.

Darker food powders, such as cocoa powder, usually express low lightness, with values of *L** 42 [45]; lightly roasted coffee presented values of *L** 26 [46]. Usually, these darker colors are perceived by consumers as having a bitter taste [10]. Well-accepted protein powders, such as whey protein, exhibit light colors, with high values of *L**, which increase from 71 to 90 [47]. This indicates that lighter powders can be useful for the food industry, as they are more accepted and allow for the addition of other pigments during the process, increasing the acceptability of the food. Therefore, the samples treated with ethanol 99.5% or hexane, which showed a higher value of *L**, might be better accepted by the final consumer. 

#### 3.5.2. Water-Holding Capacity (WHC)

The values of water-holding capacity (WHC) are shown in Figure 6.

Our study showed that in the original powder, the WHC was 1.40 g water/g powder. Consistent with our results, Mintah et al. found that the WHC for *Hermetia illucens* was 1.36 g water/g powder [48]. Another study conducted by Zielińska et al. in *Tenebrio molitor* powder was also in accordance with our data, showing a WHC of 1.29 g water/g powder [49]. As expected, WHC increased with the treatments, especially in the ethanol 99.5% treatment (*p <* 0.05), increasing to 2.30 g water/g powder. Séré et al. [13] reported that, in defatted insect powder, the WHC can range between 2.03 g water/g powder and 4.83 g water/g powder. Ndiritu et al. [28] showed that cricket powder defatted using hexane had a WHC of 2.03 g water/g powder. These studies obtained similar values to those reported in our study.

WHC is an important property since it is related to enhancing the palatability of food and changes its viscosity [48]. External factors can also affect the WHC, such as pH and temperature [50]. In our protocol, the powder treated with ethanol 99.5% was submitted to heat. Heating the powder can denature its proteins, but also improve the WHC [51]. Therefore, the increase in WHC might be explained not only by the defatting process, but also by the elevated temperature.

#### 3.5.3. Oil-Holding Capacity (OHC)

The values of OHC were shown in Figure 7. In the original powder, the OHC was low, with a value of 1.00 g oil/g powder.

OHC is the amount of fat retained by the food, usually by the proteins present in food [13]. OHC is important in food preparations since it influences the texture and sensory attributes of the food, such as flavor and taste [50]. Séré et al. showed that the lowest OHC in *C. butyrospermi* powder was 2.17 g oil/g powder [13], differring from the result obtained in our study. However, Zielińska et al. [49] showed that the OHC of *Tenebrio molitor* powder was 1.71 g oil/g powder, which was close to the WHC of our treated powders. In the powder treated with hexane, the OHC was 1.50 g oil/g powder and the value significantly increased (*p* < 0.05) in the ethanol 99.5%, to 2.20 g oil/g powder. OHC is affected by the protein processing method, and by concentration [51]. The increase of the OHC of the powder treated with ethanol 99.5% can be explained, therefore, by the increase in protein concentration after the defatting during the treatment.

#### 3.5.4. Zeta Potential and pH

Zeta potential values were expressed in Table 2. In the original powder, the zeta potential value was −27.9 mV, indicating a possible repulsion among the particles due to a high negative charge. Zeta potential can be used to verify and guarantee the stability of particles. A very high or very low zeta potential (highly charged particles) can prevent the agglomeration of the particles due to electric repulsion [52]. When compared to the work of Kingwascharapong et al. [52], who analyzed *Bombay locust zeta* potential at the same pH as our sample (pH 7), the *Gryllus bimaculatus* powder had a higher negative zeta potential, with more repulsion among the particles.

When compared to the other treatments, there was no statistical difference among the conducted treatments except for the pH 5 treatment, which presented a zeta potential of +16.2 mV.

This difference can be explained due to the pH of this powder (pH 4.9) and the fact that zeta potential is influenced by the pH of the samples [53]. According to Hiremath [53], a low pH leads to a positive zeta potential and a higher pH will lead to a negative zeta value. The isoelectric point of cricket proteins is around pH 4 [32], which might influence the zeta potential. If aiming to use the powder for 3D food printing this characteristic can be useful since the agglomeration of the particles could clog the printer nozzle [54].

#### 3.5.5. Flowability

The Carr index is a well-established method to determine the flowability of the powders. In this regard, the parameters Angle of Repose (AoR) and Cohesion were used. The values and Carr index for AoR are represented in Figure 8. It is possible to observe that, initially, the original powder had an AoR of 47.2°, classified as a ‘not good’ flowability [54] with a Carr index of 12.

Upon treatment with hexane or ethanol 99.5%, this value decreased to 37.8 and 42.5° and the index increased to 16 or 18, respectively, improving it (*p <* 0.05) to the status of ‘normal’ or ‘good’ flow, respectively [55]. The improvement in the AoR can be attributed to defatting from the powder. Groesbeck et al. [56] showed that, upon the addition of fat (soy oil) to a food powder, the AoR tends to increase (decreasing the Carr index), resulting in a bad flowability when compared to the sample without fat. Tuohy et al. [57] evaluated the AoR in a study with whole milk and skim milk powders and showed that, in the case of skim milk, the AoR significantly decreased when compared to the whole milk (with an AoR of 36° for skim milk powder, and 56° for whole milk powder), indicating that the fat can affect the powder flowability. 

Cohesion is defined as the force of inter-particle attraction that prevents the sliding of one particle to another [58]. The cohesion data in the cricket powders are expressed in Table 3. It was not possible to evaluate the cohesion of the original powder, since it was too cohesive and its particles tended to agglomerate so that the powder could not pass through the sieves. This meant that it was classified as having a ‘very bad’ flowability according to the definition of the Carr index [55]. After defatting treatments (hexane or ethanol 99.5% treatment), it was possible to measure the cohesion of the powders. The hexane-treated powder showed a cohesion of 50.3 and the ethanol 99.5% had a cohesion of 48.7, both with a Carr index of 7, improving the flowability from ‘very bad’ to ‘not good’ [55]. According to Barbosa-Cánovas et al. [58], cohesive materials will fail in terms of flow, even when flowing through an opening 1000 times larger than the size of the particles. This is especially likely with food powders rich in fat or oil.

Moisture or fat concentration can change the flowability of a powder [59]. 

Characterizing the flowability of food powders is essential to the manufacturing process. When the powder is going to be used for additive manufacturing (3D printing), evaluating the flowability can guarantee the success of the process [59]. Many factors can influence powder behavior and its flow, such as oil and moisture [60].

#### 3.5.6. Paste Properties—Viscosity

Figure 9 shows the viscosity of the paste made with the powders. The viscosities started around 375 mPa.s and decreased to around 150 mPa.s as the shear rate increased. The viscosity of all the treated powders was similar, except for the powder treated with ethanol 99.5%, which had a lower initial viscosity. According to Bourne et al. [61], suspended matter tends to increase the viscosity of the food. Since the suspended matter present in the ethanol 99.5%-treated powder was lower than in the other powders—after fat removal, likely resulting in a higher capacity to absorb water—its viscosity tends to decrease.

Viscosity is an important parameter for foreseeing food industry applications, such as in 3DFP and other forms of food production. Materials with a high viscosity of 6000 mPa.s or above can cause nozzle clogging in the 3D food printer during the printing process [62].

Smet et al. [63] evaluated the viscosity of mealworm (*Tenebrio molitor*) pastes and showed that the viscosity of this edible insect powder was similar to the one found in our study: 342 mPa.s.

As can be seen in the graph, the pastes of all the powders had a shear-thinning behavior. To classify the flow as a shear-thinning behavior, the behavior of the viscosity as a function of shear rate was observed [62]. Shear-thinning behavior was found by Pant et al. [64], in a study on the use of food inks for 3D food printing, using vegetables as ink material. The printability of these inks was classified as good since the printed material kept its shape and form after the printing process. According to Liu et al. [61], food inks must have a low viscosity and a shear thinning behavior, to be more easily extruded and keep their shape.

#### 3.5.7. Relationship between the Chemical Composition of the Powders and Visual, Techno-Functional, and Rheological Parameters

In order to show evidence of the composition of the powders and how they influence the other analyzed parameters, a correlation analysis was performed. The correlation is shown in Table 4.

The composition of the powders directly influences the powder properties, as suggested in the previous discussions. The fat content directly affects the color of the powder due to the removal of melanin [28], which is a lipophilic component [42]. Fat also influenced the WHC and OHC of the sample [13], which increased as the fat content is reduced. A similar pattern was observed when AoR was evaluated; a better flowability was found when using the defatting process. Fat directly influences all the analyzed factors, and the removal of this compound improved the powder characteristics.

Protein content also affected the powder properties, specifically the WHC and OHC [51]. Since these factors were affected by protein content in the sample, increasing the protein concentration will improve these two parameters.

The treatments applied to the powders influenced in their main composition, which directly affected the visual aspects, techno-functional, and rheological properties of the powders.

## 4. Conclusions

In conclusion, the findings of this study provide valuable insights into the potential benefits of using solvent treatments for edible insects. Using 99.5% ethanol and hexane treatments, the increase in protein concentration, decrease in fat concentration, and color reduction, coupled with the improvement in other functional properties such as OHC, WHC, and flowability, make the insect powder a highly attractive ingredient for food manufacturers. Not only could it help in the development of healthier and more sustainable food products, but it could also improve the overall quality and texture of various food products. Furthermore, the use of *Gryllus bimaculatus* powder as a food ingredient has the advantage of being a more environmentally friendly and sustainable source of protein. Overall, this study highlights the potential benefits of using solvent treated powders, especially defatted ones, as a versatile and functional ingredient for the food industry with potential applications in a wide range of food products.

## Figures and Tables

**Figure 1 foods-12-01422-f001:**
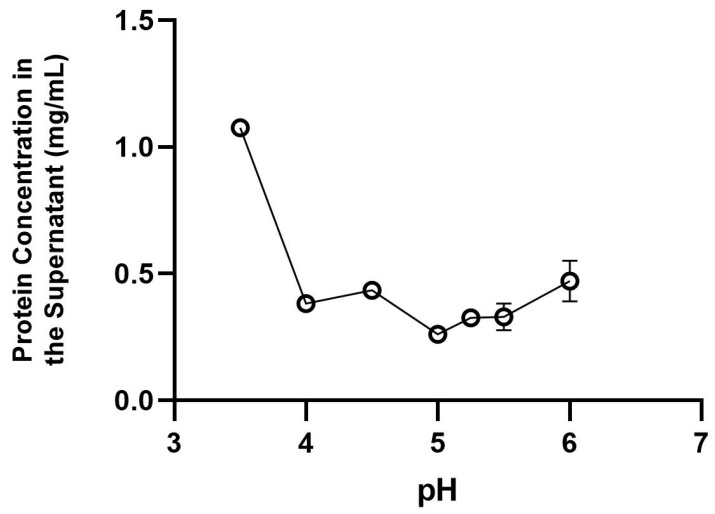
Effect of the pH conditions on protein transference from the powder to the supernatant.

**Figure 2 foods-12-01422-f002:**
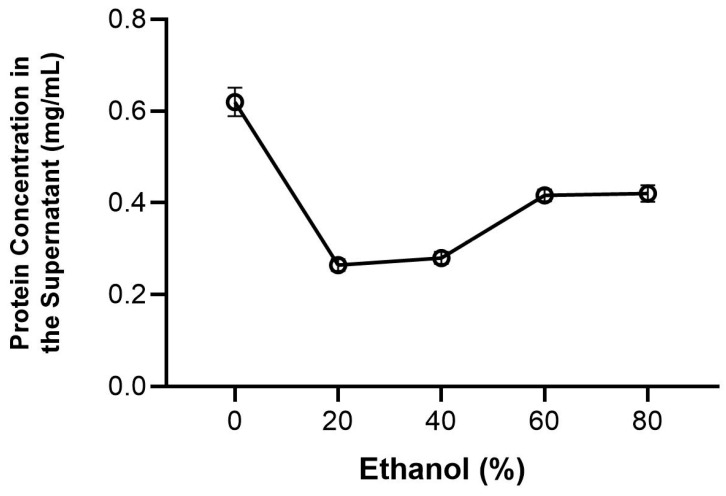
Effect of the ethanol concentration on protein transference from the powder to the supernatant.

**Figure 3 foods-12-01422-f003:**
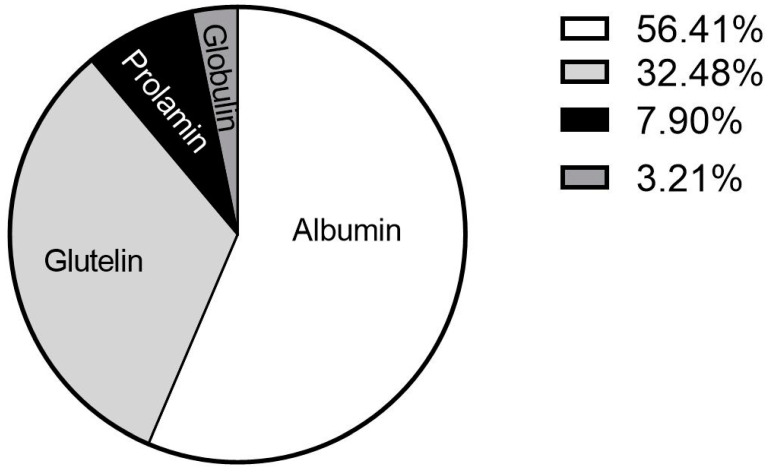
Osborne’s Classification of proteins of original powder.

**Figure 4 foods-12-01422-f004:**
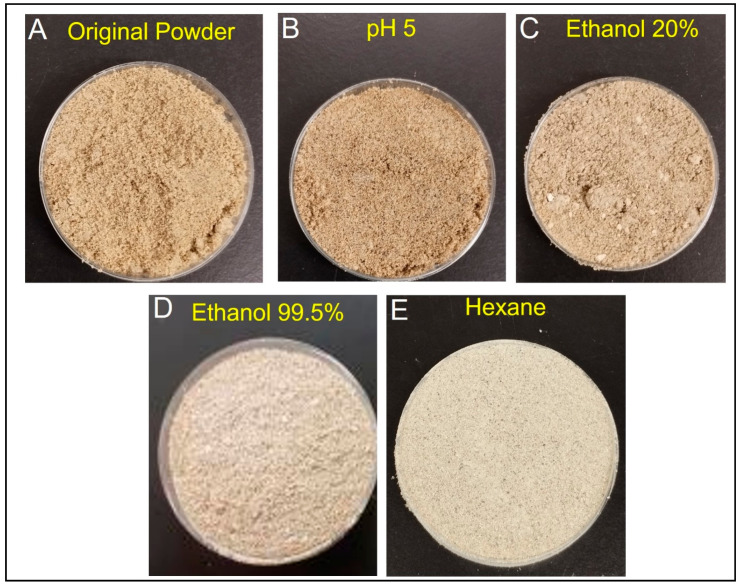
Pictures of the original powder and treated powders. (**A**) = original powder, (**B**) = pH 5 treated powder, (**C**) = ethanol 20% treated powder, (**D**) = ethanol 99.5% treated powder, (**E**) = hexane treated powder.

**Figure 5 foods-12-01422-f005:**
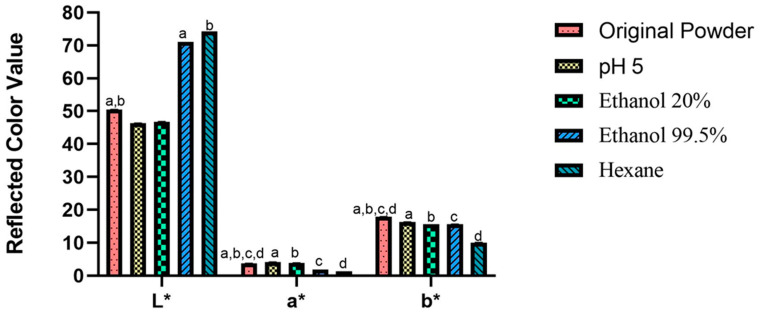
Difference in powder colors using CIELAB. Letters represent significant difference (*p <* 0.05). *L**—a = ethanol 99.5% vs. original powder, b = hexane vs. original powder. *a** and *b**—a = pH 5 vs. original powder, b = ethanol 20% vs. original powder, c = ethanol 99.5% vs. original powder, d = hexane vs. original powder.

**Figure 6 foods-12-01422-f006:**
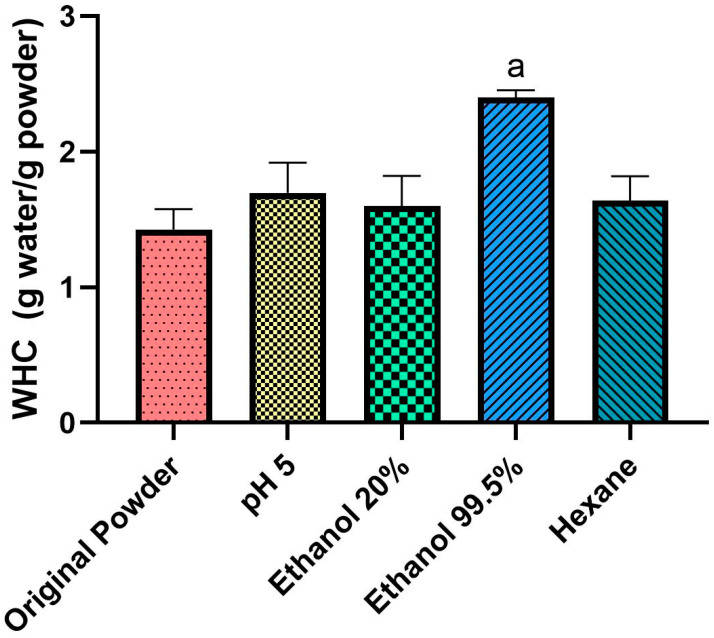
Water-holding capacity (WHC) before and after treatments in cricket powder. Letters represent significant difference (*p <* 0.05). a = ethanol 99.5% vs. all the powders.

**Figure 7 foods-12-01422-f007:**
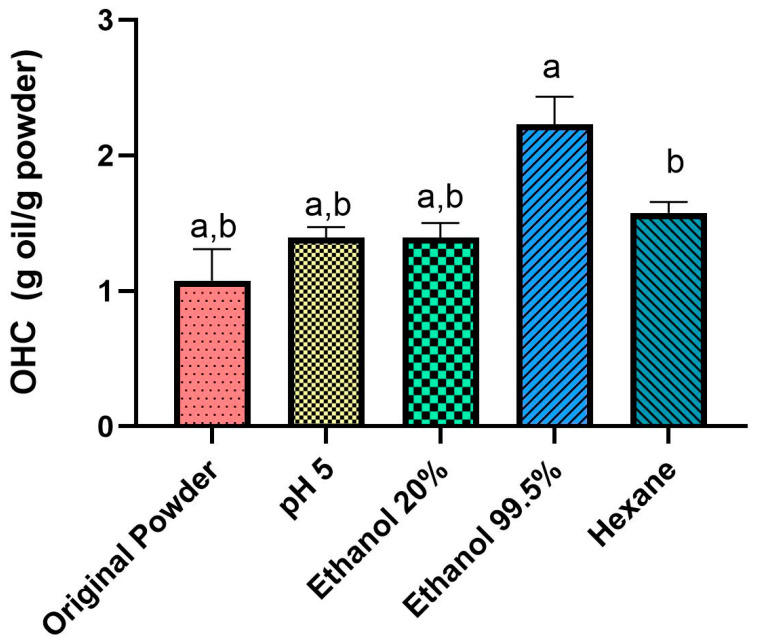
Water holding capacity (OHC) before and after treatments in cricket powder. Letters represent significant difference (*p <* 0.05). a = ethanol 99.5% vs. original powder, pH 5, and ethanol 20%. b = hexane vs. original powder, pH 5, and ethanol 20%.

**Figure 8 foods-12-01422-f008:**
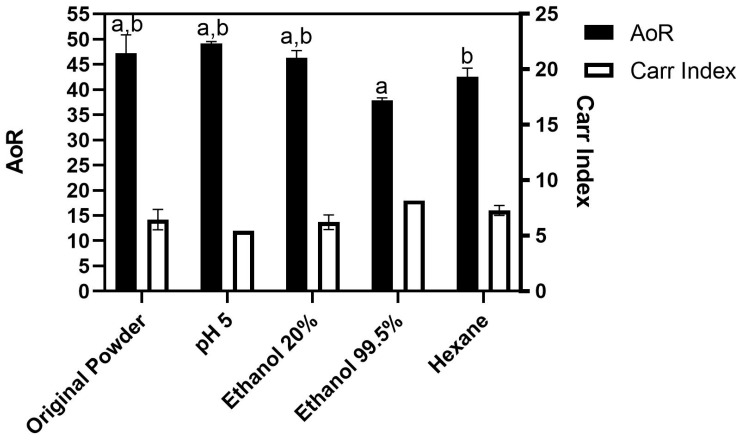
Evaluation of flowability using angle of repose (AoR). Letters represent significant difference (*p <* 0.05). a = Ethanol 99.5% vs. original powder, pH 5, and ethanol 20%. b = hexane vs. original powder, pH 5, and ethanol 20%. Carr index classification ranges from 0 to 25, in which 0–4.5 is very bad, 5–9.5 is bad, 10–14.5 is not good, 15–17 is normal, 17.5–19.5 is good, 20–22 is fairly good and 22.5–25 is very good flow. Classification of flowability according to Carr Index. (Hosokawa Powder Tester Model PT-X Operating Manual, Page D-53).

**Figure 9 foods-12-01422-f009:**
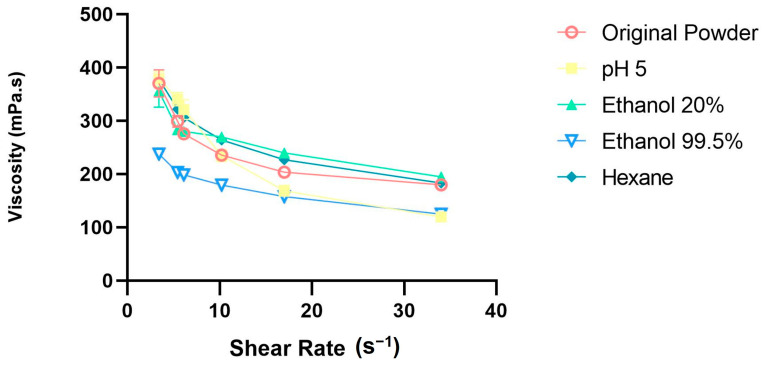
Pasting properties of cricket powders and treatments: evaluation of viscosity.

**Table 1 foods-12-01422-t001:** Proximate analysis, recovery and characterization of cricket powder and treatments.

	Original Powder	pH 5 Treated Powder	Ethanol 20% Treated Powder	Ethanol 99.5% Treated Powder	Hexane Treated Powder
Total powder recovery (%)	-	79.1 ± 2.0 ^a^	72.3 ± 1.0 ^b^	63.3 ± 2.8 ^c^	67.1 ± 0.2 ^d^
Protein (%)	55.4 ± 1.0 ^a^	56.0 ± 2.2 ^a^	60.4 ± 0.6 ^b^	72.5 ± 0.4 ^c^	68.5 ± 1.9 ^d^
Protein recovery (%)	-	79.8 ± 3.1 ^a^	78.8 ± 0.8 ^a^	82.9 ± 0.4 ^a^	83.0 ± 2.3 ^a^
Fat (%)	33.0 ± 0.6 ^a^	29.3 ± 0.2 ^b^	23.0 ± 0.7 ^c^	6.8 ± 0.3 ^d^	7.8 ± 0.4 ^e^
Fat recovery (%)	-	70.3 ± 0.4 ^a^	50.4 ± 1.5 ^b^	13.0 ± 0.6 ^c^	16.0 ± 0.9 ^d^
Chitin (%)	7.3 ± 0.2 ^a^	8.7 ± 0.9 ^b^	9.8 ± 0.8 ^b^	9.5 ± 0.5 ^b^	9.8 ± 0.8 ^b^
Chitin recovery (%)	-	90.9 ± 10 ^a^	93.5 ± 6.3 ^a^	82.6 ± 4.3 ^a^	90.6 ± 7.4 ^a^
Ash (%)	3.2 ± 0.0 ^a^	1.5 ± 0.2 ^b^	2.2 ± 0.2 ^a^	4.1 ± 0.2 ^a^	2.8 ± 0.3 ^a^
Ash recovery (%)	-	41.7 ± 0.6 ^a^	50.2 ± 5.4 ^a^	85.5 ± 1.3 ^b^	62.4 ± 5.3 ^c^
Moisture (%)	3.7 ± 0.3 ^a^	3.5 ± 0.2 ^a^	2.7 ± 0.1 ^a^	4.5 ± 0.3 ^a^	4.6 ± 0.1 ^a^
Total phenolic compounds (g/kg)	6.0 ± 0.17 ^a^	4.3 ± 1.1 ^b^	4,6 ± 1.7 ^b^	7.7 ± 3.6 ^c^	5.6 ± 3.3 ^a^
Total phenolic compounds recovery (%)	-	57.3 ± 1.5 ^a^	55.6 ± 2.0 ^a^	82.0 ± 3.8 ^b^	62.5 ± 3.7 ^a^
Tannins (g/kg)	13.0 ± 1.7 ^a^	7.3 ± 2.5 ^b^	5.9 ± 1.9 ^b^	16.0 ± 4.6 ^a^	16.0 ± 2.6 ^a^
Tannins recovery (%)	-	57.3 ± 1.5 ^a^	55.6 ± 2.0 ^a^	82.0 ± 3.8 ^b^	62.5 ± 3.7 ^a^
Phytates (mg/kg)	9.32 ± 0.08 ^a^	8.92 ± 0.08 ^b^	9.12 ± 0.05 ^a^	8.75 ± 0.09 ^a^	9.19 ± 0.36 ^a^
Phytates recovery (%)	-	75.2 ± 0.1 ^a^	70.8 ± 0.4 ^b^	59.4 ± 0.6 ^c^	66.2 ± 2.5 ^d^

Results are represented as mean ± Standard Deviation (SD). – Represents non-applicable. a–e—Values in the same row with different superscript letters significantly differ (*p <* 0.05).

**Table 2 foods-12-01422-t002:** pH and zeta potential of the powders.

	Original Powder	pH 5 Treated Powder	Ethanol 20% Treated Powder	Ethanol 99.5% Treated Powder	Hexane Treated Powder
pH	7.0 ^a^	4.9 ^b^	7.3 ^a^	7.0 ^a^	6.8 ^a^
Zeta potential (mV)	−27.9 (−33.8–−25.8)	16.2 (15.4–17.0)	−31.5 (−31.5–−30.4)	−19.25 (−17.9–−20.3)	−27.5 (−28.0–−26.9)

Results are represented as mean and percentiles. a, b: values in the same row with different superscript letters differ significantly (*p <* 0.05).

**Table 3 foods-12-01422-t003:** Cohesion and Carr index of original and treated cricket powder.

Material	Cohesion	Carr Index
Original Powder	n/a	0
pH 5	n/a	0
Ethanol 20%	59.8 ± 2.6	2
Ethanol 99.5%	50.3 ± 3.5	7
Hexane	48.7 ± 1.0	7

n/a = Not applicable, as the sample was too cohesive and measuring the cohesion was not possible. Results are represented as mean ± Standard Deviation (SD). Carr index classification ranges from 0 to 25, in which 0–4.5 is very bad, 5–9.5 is bad, 10–14.5 is not good, 15–17 is normal, 17.5–19.5 is good, 20–22 is fairly good and 22.5–25 is very good flow. Classification of flowability according to Carr Index. (Hosokawa Powder Tester Model PT-X Operating Manual, Page D-53).

**Table 4 foods-12-01422-t004:** Correlation between the powder composition and powder properties.

Comparison	Correlation?	*p* Value
Fat × Color	Yes	<0.01
Fat × WHC	Yes	<0.01
Fat × OHC	Yes	<0.01
Fat × AoR	Yes	<0.01
Protein × WHC	Yes	<0.01
Protein × OHC	Yes	<0.01

Correlation between the factors analyzed by Pearson’s Correlation analysis. All the correlations were positive.

## Data Availability

The data presented in this study are available on request from the corresponding author.

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
