# Peer review of "Improvements in Visual Aspects and Chemical, Techno-Functional and Rheological Characteristics of Cricket Powder (Gryllus bimaculatus) by Solvent Treatment for Food Utilization"

_foods, 2023, doi:10.3390/foods12071422_

Round 1

Reviewer 1 Report

The study explores improving cricket powder's physical and chemical characteristics for food utilization. 

- Overall the manuscript is well-written, but requires minor corrections.  

Some of the corrections or suggestions are as below:

1. Title is given as only physical and chemical characteristics, it can be modified as other properties are also measured such as viscosity, SEM, WHC, OAC etc..... Modify the title. 

2. Abstract is not clear, it has to rewritten, be more specific. Like Line18-20, it is not clear..... increase in what treatment etc has to be mentioned clearly.  Line 21-22 - Not clear.  Line 18 - it is physical and chemical.

2.  Line 37 - m3- superscript- check the whole manuscript for corrections

3. In Material and Method- Add a subheading on statistical methods- details on method used, software used etc

4. Line 279- pro-tein, write as protein.

Author Response

RESPONSES TO REVIEWER 1

Open Review

Quality of English Language

( ) English very difficult to understand/incomprehensible
( ) Extensive editing of English language and style required
( ) Moderate English changes required
(x) English language and style are fine/minor spell check required
( ) I am not qualified to assess the quality of English in this paper

Yes

Can be improved

Must be improved

Not applicable

Does the introduction provide sufficient background and include all relevant references?

(x)

( )

( )

( )

Are all the cited references relevant to the research?

(x)

( )

( )

( )

Is the research design appropriate?

(x)

( )

( )

( )

Are the methods adequately described?

(x)

( )

( )

( )

Are the results clearly presented?

(x)

( )

( )

( )

Are the conclusions supported by the results?

(x)

( )

( )

( )

Comments and Suggestions for Authors

The study explores improving cricket powder's physical and chemical characteristics for food utilization. 

- Overall the manuscript is well-written, but requires minor corrections.  

Response: The authors appreciate your kind reviews and the effort and time applied to reviewing this manuscript. We are grateful for all your feedback, which will help us to improve our work.

Some of the corrections or suggestions are as below:

  1. Title is given as only physical and chemical characteristics, it can be modified as other properties are also measured such as viscosity, SEM, WHC, OAC etc..... Modify the title. 

Response: We appreciate your suggestion regarding the title. We all agree that the title could contain more specific information about what was done in the study. As suggested, we modified the title to “Improvement of Visual Aspects, Chemical, Techno-Functional and Rheological Characteristics of Cricket Powder (Gryllus bimaculatus) by Solvent-Treatment for Food Utilization”, which includes all the analysis performed during the study.

  1. Abstract is not clear, it has to rewritten, be more specific. Like Line 18-20, it is not clear..... increase in what treatment etc has to be mentioned clearly.  Line 21-22 - Not clear.  Line 18 - it is physical and chemical.

Response: We would like to thank the reviewer for the comment on the manuscript’s abstract. It was rewritten as suggested, following all revisor’s suggestions. The points not clear were explained in the abstract, including the treatments that resulted in each result explained. The new abstract follows in Lines: 14-27

  1. Line 37 - m3- superscript- check the whole manuscript for corrections

Response: The authors really appreciate the reviewer’s efforts and attention to the details when revising the manuscript. We reviewed the whole manuscript and corrected the typing mistakes, including the superscript of m3.

  1. In Material and Method- Add a subheading on statistical methods- details on method used, software used etc

Response: We acknowledge the reviewer's suggestion. Following your kind suggestion, we included the subheading “2.20. Statistical Analysis” in the Lines: 268-275 , and described the methods used, the number of samples, the significant difference chosen, and the software used information, as follows:

            “For the statistical analysis, all the measurements were performed in triplicates. The statistical test used for analysis was the one-way analysis of variance (ANOVA) with the Holm-Sidak as post hoc. Also, to evaluate the correlation between the powder composition and powder properties, Pearson’s Correlation test was performed. All the analyses were performed using SPSS statistics software, version 29.0 (IBM Corp., Armonk, NY,USA). The results were expressed as mean and standard deviation. The significant difference was considered at p < 0.05”

  1. Line 279- pro-tein, write as protein.

Response: Once again, the authors are grateful to the reviewer for the attention to details in our manuscript. It will help us to improve it. The typing mistake of the word protein was corrected in to protein.

Reviewer 2 Report

Interesting study, manuscript and outcomes. However, other novel extraction technologies could be used in line with the objective of authors reducing GHG and contributing to defining other protein sources. However, today consumers wishing to avoid any hexane-extracted foods. In food industry, regulations and marketing authorizations are limited regarding the extraction solvents for food additives.  

The manuscript under revision discusses the utilization of different solvents for extraction of protein from cricket powder with the objective of increasing its acceptance for use as a protein source in food production. Researchers have used solvents such as ethanol and hexane.

Since the end product is aimed to be introduced as a food ingredient the extraction procedure should be food grade, GRAS and safe.

How likely the hexane can have green pass?

“These findings suggest that cricket powder treated with solvents can be used as a protein source in different food applications and may help increase the acceptance of edible insects as viable protein sources.”

Insect proteins may eventually get accepted as a food ingredient however the selected extraction methods, safety, nutritional and organoleptical values are the main issues which require investigations.

[Line 58-60] The taste (important organoleptic characteristic) of insect proteins is basically unknown to so many. Why researchers didn’t choose this aspect to investigate, as well.

Author Response

RESPONSES TO REVIEWER 2

Open Review

Quality of English Language

( ) English very difficult to understand/incomprehensible
( ) Extensive editing of English language and style required
(x) Moderate English changes required
( ) English language and style are fine/minor spell check required
( ) I am not qualified to assess the quality of English in this paper

Yes

Can be improved

Must be improved

Not applicable

Does the introduction provide sufficient background and include all relevant references?

( )

(x)

( )

( )

Are all the cited references relevant to the research?

(x)

( )

( )

( )

Is the research design appropriate?

( )

(x)

( )

( )

Are the methods adequately described?

(x)

( )

( )

( )

Are the results clearly presented?

(x)

( )

( )

( )

Are the conclusions supported by the results?

( )

(x)

( )

( )

Comments and Suggestions for Authors

Interesting study, manuscript and outcomes. However, other novel extraction technologies could be used in line with the objective of authors reducing GHG and contributing to defining other protein sources. However, today consumers wishing to avoid any hexane-extracted foods. In food industry, regulations and marketing authorizations are limited regarding the extraction solvents for food additives.

The manuscript under revision discusses the utilization of different solvents for extraction of protein from cricket powder with the objective of increasing its acceptance for use as a protein source in food production. Researchers have used solvents such as ethanol and hexane.

Since the end product is aimed to be introduced as a food ingredient the extraction procedure should be food grade, GRAS and safe.

Response: The authors would like to express their sincere gratitude to the reviewer for all the suggestions and comments, which will allow us to improve our manuscript. Regarding the use of other extraction technologies, the authors agree that other methods could also be used, but in the first moment, the use of solvents was chosen to be analyzed. Also, the main objective of the study is to improve the characteristics of the powders, but not specifically to reduce GHG emissions. The use of edible insects is responsible for reducing GHG emissions when compared to traditional livestock.

The use of hexane in the food industry is common. One example is the extraction of vegetable oil, especially soybean oil, which uses hexane as a solvent. Solvent extraction of vegetable oils is the most common and efficient method for oil production.  According to the European Directive 2009/32/EC, to prepare defatted protein products and defatted flours, the accepted levels of hexane for the final consumer range from 10 to 30 mg/kg. These concentrations of hexane in food are considered safe to be ingested by the final consumer.

Hexane has a high vapor pressure (153 mm Hg at 25 °C), which makes this solvent highly volatile at room temperature. Also, it is a fast-evaporating liquid, which indicates that, after the drying process in the fume hood and in the freeze-drying equipment, almost all hexane used for treatment must be eliminated/ dried.

How likely the hexane can have green pass?

Response: The authors agree that this is a very relevant and important question, and we would like to express our gratitude for bringing that point. We agree that hexane cannot have a green pass, but techniques to reuse this solvent can be applied, avoiding the unnecessary discard of the solvent and a better and “greener” utilization can be done.

“These findings suggest that cricket powder treated with solvents can be used as a protein source in different food applications and may help increase the acceptance of edible insects as viable protein sources.”

Insect proteins may eventually get accepted as a food ingredient however the selected extraction methods, safety, nutritional and organoleptical values are the main issues which require investigations.

Response:  According to FAO,2021, the safety of edible insects has been ensured, and their consumption by humans is considered safe. The safety of the treatments can be secured by removing completely the solvents from the sample. Also, we used hexane as a positive control for this comparative study since it is able to remove fat and hydrophobic substances. The target treatments would be the aqueous ones and the ethanol 99.5%, which has similar results when compared to hexane.

The nutritional value of the samples was presented in the proximate analysis, where we described the protein, fat, chitin, and ashes content (in Table 1, Lines: 347-375). Also, phenolic compounds and anti-nutrients are presented in our study, again represented in Table 1. The organoleptic analysis must be conducted in future studies using cricket powder.

[Line 58-60] The taste (important organoleptic characteristic) of insect proteins is basically unknown to so many. Why researchers didn’t choose this aspect to investigate, as well.

Response: The authors appreciate the suggestion of the reviewer and agree with the importance of the evaluation of the taste and other organoleptic characteristics. This aspect will be investigated in future studies since for this paper, the main objective was to characterize the cricket powder and improve visual aspects, chemical, techno-functional, and rheological properties.

Reviewer 3 Report

Dear authors

My suggestions and recommendations are inserted in the attached file

Regards

Author Response

RESPONSES TO REVIEWER 3

Open Review

Quality of English Language

( ) English very difficult to understand/incomprehensible
(x) Extensive editing of English language and style required
( ) Moderate English changes required
( ) English language and style are fine/minor spell check required
( ) I am not qualified to assess the quality of English in this paper

Yes

Can be improved

Must be improved

Not applicable

Does the introduction provide sufficient background and include all relevant references?

( )

(x)

( )

( )

Are all the cited references relevant to the research?

( )

(x)

( )

( )

Is the research design appropriate?

( )

( )

(x)

( )

Are the methods adequately described?

( )

( )

(x)

( )

Are the results clearly presented?

( )

( )

(x)

( )

Are the conclusions supported by the results?

( )

(x)

( )

( )

Comments and Suggestions for Authors

Dear authors

My suggestions and recommendations are inserted in the attached file

Regards

Response: We would like to express our gratitude to the reviewer for kindly spending your valuable time on a thorough review of our manuscript. All the comments and suggestions will help us to improve our work. Also, we would like to deeply apologize due to, some changes could not be done due to a possible conflict with the previous reviews.

Regarding the English language, all the authors reviewed the manuscript and the corrections were done accordingly.  

The authors answered comments from the reviewer in the order they appear in the text.  

  • Which treatment?

Response: The authors appreciate your question in the abstract. The treatments are mentioned in the following sentence, due to the limited words in the abstract, we only mentioned them in the methodology description of the abstract, in Lines: 16-17. The treatments are done using the following solvents: pH 5 aqueous solution, ethanol 20%, ethanol 99.5%, and hexane.

  • Be careful about acceptance. The acceptance is linked with sensory consumer traits.

Response: We are so grateful to the reviewer for pointing out this issue. The word acceptance was removed from the abstract as suggested by the reviewer to avoid any misunderstanding about the main objective of our work.

  • Physical and chemical.

Response: Thank you so much for your detailed revision and for noticing this point. The word was corrected to just chemical (Line: 14), due to the modifications suggested by the other reviewers in the title and consequently in the abstract.

  • Accord these value on %. Authors should compare all the data

Response: The authors really appreciate your suggestion regarding changing the proximate analysis (protein, chitin, fat, moisture and ash) from g/100 g to %. We have changed the unit as solicited by the reviewer in the whole manuscript, including in the abstract where it was suggested. The changes were done in Lines: 19-20, 336-338, 341-342, 347-365, 384-385, 399-400, 406-407.

  • More develop the evolution of color parameters. Make a simple discussion.

Response: We would like to thank the reviewer for the insightful comment and suggestion. A simple explanation about the reason for the color change was added to the abstract, due to the limitation of words, in Line:.22-23. However, the whole explanation is given in the “Results and Discussion” section, in Lines: 468-472 , as follows:

The brownish color present in Gryllus bimaculatus powder is associated with melanin expression in this insect [28]. Due to the fact that melanin is a lipophilic compound present in the skin of animals [42], the removal of fat from the samples also removes melanin as well. This explains the lighter color resulted from hexane and ethanol 99.5% treated powders.”

  • How about the link between all the data? Physical, chemical, color, nutritional and techno-functional properties?

Response: A brief explanation about the link between all the data was added to the abstract, due to the limit of words in Lines 24-26. During the Discussion, the influence of the main composition of the samples in the analyzed parameters is explained individually. For example, in Lines: 468-472, 477-478, 518-519,539-540, 576-577,594-595.

The chemical parameters, such as fat content and protein content of the samples influence the color, nutritional value, techno-functional and rheological properties of the samples. A Pearson Correlation test was performed and there is a correlation between the fat or protein concentration in the parameters listed above. For example, the color becomes lighter (L* increased) while the fat concentration decreased in the sample.

Also, a new subhead was added to the main text to explain, as suggested by the reviewer, the correlation between the data, in Lines: 658-687  as follows:

            “ In order to show evidence of the composition of the powders and how it influences other parameters analyzed, a correlation analysis was performed. The correlation is shown in Table 4.

The composition of the powders will influence in the powder influenced the powder properties, as suggested in the previous discussions. The fat content will directly affect the color of the powder, due to the removal of melanin [28], which is a lipophilic component [42]. Fat also influenced the WHC and OHC of the sample [13], being increased as the fat content is reduced. A similar pattern is observed when AoR was evaluated, a better flowability was found with the defatting process. Fat influences directly all these factors analyzed, and the removal of this compound improved the powder characteristics.

Protein content also affected the powder properties, specifically the WHC and OHC [51]. Since these factors are affected by protein content in the sample, increasing the concentration of protein will improve these two parameters.

The treatments applied in the powders influenced the main composition of it, which directly affected the visual aspects, techno-functional and rheological properties of the powders.”

  • Please develop more this sentence. This is a critical point about the consumption of these molecules.

Response: The authors would like to deeply apologize, but we could not understand the suggestion given in this part of the manuscript.

  • What kind of treatments? Processing used for these compounds.

Response: Once again, we want to apologize to the reviewer, but the authors could not understand the requested modification/ suggestion made.

  • What is the practical application of these molecules in food industry? Limitations and technology incorporation

Response: We agree that this topic is important, and we appreciate your question. Bioactive compounds are components of food that have biological activity in the body after ingestion. The consumption of this compound is important for improving the health of consumers since it has antioxidant and other beneficial activities. For the food industry, it can be applied to the formulation of new foods, which could attract more consumers that are seeking healthier foods. The principal limitation is the extraction method of these compounds from the food. A large number of techniques can be applied and the one that fits better must be carefully chosen. The bioactive compounds can be incorporated into different types of food, such as beverages, sauces, and bakery products.

  • Rewrite this sentence

Response: We deeply appreciate the reviewer’s suggestion to improve the sentence. The sentence was rewritten as a suggestion given by the reviewer, in the Line: 58-59  as follows:

“The color of the powder can also contribute to increasing the acceptability of this new food.”

  • What the link of this sentence with this study

Response: We sincerely appreciate your question about this topic in our manuscript. The 3DFP was explained due to the possibility of future applications using edible insects. It could attract new consumers if edible insects were applied for 3DFP or other food processing. The authors also intend to use the cricket powder in future studies for 3DFP applications, and the parameters analyzed in this manuscript influence the food printing process.

  • How about the extractive solvents? The polarity of these compounds? The analytic chemistry used to characterize these molecules.

Response: The authors would like to thank the author for the question. The polarity of the solvents used is mentioned in the Results and Discussion section in Lines: 385-387  since we agree it fits better in this section. As follows:

      “The decrease of fat concentration is related to the polarity of the solvent used, non-polar solvents such as hexane, which will extract the fat from the samples.”

And more information was added in Line: 410

      “when compared with the samples treated with aqueous solvents (polar solvents).”

The used solvents for this manuscript can be divided into polar and non-polar solvents.

Hexane is characterized as a non-polar solvent because it has a symmetrical shape and its distribution of electrons within the molecule is uniform.

Ethanol is a polar solvent, due to its hydroxyl group, which is a polar functional group. Despite of that, ethanol can extract fat from the sample due to its ability to form hydrogen bonds with the lipids.

Water at pH 5 is also a polar solvent, but in this case with no ability to remove fat from the sample. However, it is efficient in removing antinutritional compounds, since it is water-soluble.

  • How about biodegradability (in human) of insects protein?

Response: The authors are thankful for your suggestion about biodegradability. We have added an explanation about the digestibility of insect proteins to the introduction section, in Lines: 75-76  as follows:

      “Séré [13] showed an increase in protein concentration and enhancement in functional parameters after solvent treatment of insect powders, such as digestibility improvement, which initially was around 82% and improved to 88% after powder treatment.”

The biodegradability or digestibility of edible insect proteins can change from one species to another. Previous studies showed that species such as Carbulla marginella have 82% of protein digestibility, while species such as Liometopum apiculatum can have a protein digestibility of 93.9% (in vitro digestion simulation). The processing methods of the edible insects can also affect their digestibility, especially cooking them.

  • The objective should be rewritten

Response: Thank you for your kind suggestion. The objective was rewritten according to the suggestion of the reviewer. We added all the analyzed parameters to the objective, in order to make it more specific in the Lines: 76-79 , as follow:

      Therefore, the main objective of this study is to characterize and improve the visual aspects, chemical, techno-functional and rheological properties of Gryllus bimaculatus powder by using solvent treatments and foreseeing food utilization.”

  • Change ethanol by EtOH in main MS

Response: We sincerely appreciate all your valuable comments and suggestions, but the authors would rather write “ethanol” in full, for the sake of the readership, in order to guarantee that all could understand the context, rather than using the abbreviation.

  • Please explain the choice of as pH (5) and the studied solvents (EtOH and water). I think it is better to use acetone as a proper solvent to extract the proteins?

Response: The authors agree that this is an important topic and would like to express our gratitude for the comment. The choice of the solvents was based on the pre-treatment we performed in item “2.1. Effect of the pH and Ethanol Concentration on Protein Behavior” Line: Summarizing, we evaluated the best pH and best ethanol concentrations to treat the powder without removing the protein from it to the supernatant. The main idea is to keep the protein in the powder and not extract it, as follows in Lines: 286-293.

      “When protein concentration was analyzed on pH 5 in this study, it showed low protein concentration (p <0.05) in the supernatant (0.26 mg/mL), indicating that this level of pH might be the best one to be used as an aqueous solvent for treating the cricket powder, since most of the protein concentration will remain in the powder and will not be lost gone to the supernatant.”

And in Lines: 299-302

      “When we analyzed the protein concentration in the supernatant as a function of the ethanol concentration, ethanol 20% was the best option to be used as a treatment, since it showed the lowest concentration of protein in the supernatant. With these data the ethanol 20% condition was chosen as a solvent for treatment in the following experiments.”

And in Lines:  305-307

      “Ethanol 99.5% was also chosen to be used as a treatment once it is efficient in removing fat from the sample and in decolorization of the powder, as discussed in further sections (3.5.1.).”

Also, we chose hexane as a control solvent to evaluate the effects of fat removal in the powder, once defatting has a beneficial effect in the samples, such as improvement of WHC, OHC, color, and flowability.

Since the main objective is to increase the protein concentration in the powder, not extract or purify or isolate it, we did not choose acetone as a solvent. However, once again the authors would like to thank the reviewer’s suggestion.

  • The illuminant, standard observer and aperture must be given. L*, a* and b* needs to be in italic. This is outlined in the guide to authors. What is the blooming time?

Response: Thank you for pointing this out. The solicited information is, indeed, very important. The information required was added to the manuscript in the Lines:116-117. The illuminant used was D65, the standard observer of 10°, and the measurement time was 1 s. The L*,a*,b* were edited to italic.

  • All the subsections should be concise.

Response: Thank you so much for your comment regarding the subsections. After reviewing it, the authors consider all the subsections concise.

  • How was the pH meter calibrated?

Response: The authors appreciate the reviewer’s concern about the calibration of the equipment. It is essential to guarantee the quality of the results. The pH meter was calibrated every day before its use following the fabricant manual. The calibration of the equipment was done using pH buffers in the pHs 4,7 and 9.

  • As recommended in the abstract all the data should be linked: all the responses and all trials. Can authors use a proper statistic tool as PCA, HCA, MLR, Pearson… This remark is crucial for paper acceptance .

Response: The authors are extremely grateful for your suggestion. As suggested, we linked the data of this manuscript using statistical analysis suggested by the reviewer (Pearson’s Correlation). We compared the correlation between the composition of the powder (which is affected by the treatments) and the powder’s properties such as color, WHC, OHC and flowability. Due to that, a new subsection was created, in Lines: 658-687 , named “3.5.8. Relationship Between Chemical Composition of the Powders and Visual, Techno-Functional, and Rheological Parameters”, as mentioned in previous responses.

  • I think it is better to link the pH with techno-functional properties no in protein concentration.

Response: Thanks for your suggestion. In this result and discussion topic (3.1.) Line:277 we explained the pre-treatment we performed to choose the pH and ethanol concentration to be used. The protein concentration here is in the supernatant, which could represent a loss of protein after the treatment. The main concept is to have a lower concentration of protein in the supernatant in order to keep this compound in the treated powder.

  • P should be in capital and italic

Response: We appreciate your attention to all the details in the manuscript. We modified all the P to italic in the manuscript.

  • P< or < ? how about the statistical tool used in the study (ANOVA, Tukey test…?)

Response: Thank you for your detailed review and for noticing this issue. The symbol was correct to < instead of > in Line:

 The statistical analysis was described as requested by the reviewers in the section “2.20. Statistical Analysis” in Lines: 268-275. The statistical test used for analysis was the one-way analysis of variance (ANOVA). All the analyzes were performed in triplicate. Also, as requested by the reviewer, a Pearson’s Correlation test was performed to link the data of the manuscript.

  • All data should be linked in this study. Please see my recommendation in abstract and MM section.

Response: The authors would like to express gratitude for all the suggestions made by the reviewer. As we mentioned previously, a new subsection was added to the main text of the manuscript to describe the link between the data of the study. The topic “3.5.8. Relationship Between Chemical Composition of the Powders and Visual, Techno-Functional, and Rheological Parameters”, was explained in question number 6.

  • Check all the superscripts and make a good discussion comparing your study with other ones.

Response: The authors appreciate the reviewer’s suggestions. Following it, we have done a discussion including a* and b* as well. The new discussion is presented in Lines: 472-478, as follows:

“Also, for a* and b* it is possible to observe a reduction of these parameters (p < 0.05). For a* the reduction is from 3.7 in the original powder to 1.3 in the hexane-treated powder. For b* the reduction occurs from 17.9 (original powder) to 10.0 (hexane-treated powder). A similar pattern is observed by Delgado-García et al., [43] when defatted amaranth flour is compared with full-fat one. Due to removal of fat, in lipophilic compounds such as pigments are removed from the sample, a* decreased from 2.3 to 1.1 and b* from 19.2 to 15.8 [43].”

Round 2

Reviewer 3 Report

accept as it is

Author Response

Response: The authors would like to thank the reviewer for the comment on the manuscript and for all the effort and time applied to reviewing it. All the previous comments and suggestions were very important to improve the manuscript. Thank you for accepting it.